# Cardiac Disease Alters Myocardial Tissue Levels of Epoxyeicosatrienoic Acids and Key Proteins Involved in Their Biosynthesis and Degradation

**DOI:** 10.3390/ijms232012433

**Published:** 2022-10-17

**Authors:** Theresa Aliwarga, Jean C. Dinh, Scott Heyward, Bhagwat Prasad, Sina A. Gharib, Rozenn N. Lemaitre, Nona Sotoodehnia, Rheem A. Totah

**Affiliations:** 1Department of Medicinal Chemistry, University of Washington, Seattle, WA 98195, USA; 2Certara/SimCYP Ltd., Sheffield S1 2BJ, UK; 3BioIVT, Baltimore, MD 21227, USA; 4Department of Pharmaceutical Sciences, Washington State University, Spokane, WA 99202, USA; 5Computational Medicinal Core, Center for Lung Biology, Division of Pulmonary, Critical Care and Sleep Medicine, Department of Medicine, University of Washington, Seattle, WA 98104, USA; 6Cardiovascular Health Research Unit, Department of Medicine, University of Washington, Seattle, WA 98101, USA; 7Division of Cardiology, University of Washington, Seattle, WA 98101, USA

**Keywords:** EETs, *cis*-EET, *trans*-EET, CYP2J2, Cytochrome P450, POR, EPHX2, CVD, arachidonic acid, proteomics, cardiac tissue

## Abstract

CYP2J2 is the main epoxygenase in the heart that is responsible for oxidizing arachidonic acid to *cis*-epoxyeicosatrienoic acids (EETs). Once formed, EETs can then be hydrolyzed by soluble epoxide hydrolase (sEH, encoded by *EPHX2*) or re-esterified back to the membrane. EETs have several cardioprotective properties and higher levels are usually associated with better cardiac outcomes/prognosis. This study investigates how cardiovascular disease (CVD) can influence total EET levels by altering protein expression and activity of enzymes involved in their biosynthesis and degradation. Diseased ventricular cardiac tissues were collected from patients receiving Left Ventricular Assist Device (LVAD) or heart transplants and compared to ventricular tissue from controls free of CVD. EETs, and enzymes involved in EETs biosynthesis and degradation, were measured using mass spectrometric assays. Terfenadine hydroxylation was used to probe CYP2J2 activity. Significantly higher *cis*- and *trans*-EET levels were observed in control cardiac tissue (*n* = 17) relative to diseased tissue (*n* = 24). Control cardiac tissue had higher CYP2J2 protein levels, which resulted in higher rate of terfenadine hydroxylation, compared to diseased cardiac tissues. In addition, levels of both NADPH-Cytochrome P450 oxidoreductase (POR) and sEH proteins were significantly higher in control versus diseased cardiac tissue. Overall, alterations in protein and activity of enzymes involved in the biosynthesis and degradation of EETs provide a mechanistic understanding for decreased EET levels in diseased tissues.

## 1. Introduction

Cardiovascular disease (CVD) remains the leading cause of mortality worldwide. The World Health Organization reported that 32% of global deaths in 2019 were due to CVD. Of these deaths, approximately 13% were caused by coronary artery disease (CAD) [1]. Several key risk factors, that include underlying medical conditions and lifestyle choices, contribute to the pathogenesis of CVD. In addition, inflammation has been implicated in the pathogenesis, progression, and manifestation of CVD [2,3].

Epoxyeicosatrienoic acids (EETs) are autocrine and paracrine fatty acid mediators in the cardiovascular system and are involved in modulation of vascular tone, anti-inflammatory properties, migratory and proliferative of vascular endothelial and smooth muscle cells, and anti-platelet aggregation properties. The major enzymes, and enzyme families, involved in formation of endogenous EETs are Cytochrome P450 (CYP) epoxygenases and phospholipase A_2_ (PLA_2_). EETs are formed through CYP-mediated metabolism of arachidonic acid (AA). AA is usually found esterified at the *sn*-2 position of phospholipids in cellular membranes. AA must be released from cellular membranes by PLA_2_ prior to metabolism CYP epoxygenases. Notably, AA can be metabolized by several CYP isoforms, however, the CYP2 family members, e.g., CYP2J2 and CYP2C members, are likely the most responsible for metabolism of AA. There are several classes of PLA_2_ enzymes, but calcium dependent cytosolic PLA_2_ type IVA (cPLA_2_𝑎), encoded by *PLA2G4A*, is considered the key isozyme responsible for hydrolyzing AA from the membrane [4,5]. Similar to AA, EETs are also found esterified in the membrane and saponification or hydrolysis by PLA_2_ is required to release them from this site [6,7,8]. Within the CYP catalytic cycle, there are redox partners that are crucial to carry out the rate-limiting two electron transfer steps from the cofactor, NADPH to the heme iron center. It is well established that the first electron transfer from NADPH, which reduces the ferric heme to ferrous, is catalyzed by NADPH-Cytochrome P450 oxidoreductase (POR). Therefore, the rate of CYP mediated formation of EETs will depend, in part, on the expression and activity of POR in cardiac tissue.

The degradation of EETs must also be considered when evaluating endogenous EET levels. Once formed, EETs are short lived and can be rapidly hydrolyzed to less biologically active dihydroxyeicosatrienoic acids (DHETs) by soluble epoxide hydrolase (sEH), which is encoded by the gene *EPHX2* [9]. Most EETs are re-esterified back into the phospholipid membrane for storage by acyl CoA synthase(s) [10]. It is unclear which isoform(s) of acyl CoA synthase are responsible for the re-esterification of AA and EETs. However, based on a single nucleotide polymorphism association study in sudden cardiac arrest (SCA), *LPCAT1* and *PLA2G4A* were two of the few genes that were associated with incidence of SCA [11]. *LPCAT1* encodes for lysophosphotidylcholine acyltransferase 1 that is found mainly in phosphatidylcholine lipid droplets [12]. LPCAT1 was shown to be protective against cytotoxicity induced by excess polyunsaturated fatty acids in HeLa cells by regulating the production of dipalmitoylphosphatidylcholine [13].

Numerous disease states can alter the expression of enzymes involved in the biosynthesis and degradation of EET and consequently alter steady-state EET levels. Significantly higher total EETs were reported in both obese and non-obese CAD patients compared to healthy volunteers [14]. While a follow up study by the same group reported that there were significantly lower levels of total plasma EETs in obstructive CAD patients compared to healthy volunteers [15]. The authors suggested that the contradictory results from their studies were due to significant differences in sEH activity, as indicated by altered ratios of 14,15-EET to 14,15-DHET. In addition, another group observed that plasma levels of EETs were also significantly lower in subjects with renovascular disease compared to control subjects [16]. Transcripts of *EPHX2*, the gene that encodes sEH were significantly lower in ischemic human heart failure patients compared to controls [17]. Another mechanism by which disease state impacts steady-state EET levels could be through HMOX1. Disease state increases expression of the stress response protein, heme oxygenase 1 encoded by *HMOX1* [18]. Previous work from our lab demonstrated that *HMOX1* gene expression was elevated in adult ventricular myocytes after exposure to exogenous *cis*-EETs [19]. While the relationship between *cis*-EET levels and HMOX1 has yet to be elucidated, we chose to evaluate this protein as a possible secondary marker of disease. Multiple animal studies have reported the upregulation of HMOX1 in atherosclerosis and myocardial infarction (MI) [20,21]. In addition, lipoprotein-associated PLA_2_ (Lp-PLA_2_) encoded by *PLA2G7* is abundant in coronary atherosclerotic plaques and elevated levels of this enzyme are associated with coronary heart disease events in healthy older adults [22,23]. However, studies on the mechanism(s) by which POR, cPLA_2_𝑎, and LPCAT1 are regulated in CVD are scarce. Specifically, determining how heart disease alters the expression of these proteins in human cardiac tissue can provide unique insights into potential mechanisms in this relevant organ. Since CYP2J2, POR, sEH, cPLA_2_𝑎, and LPCAT1 are all involved in EETs biosynthesis and biodegradation, as summarized in Figure 1, it is important to investigate how CVD affects their protein expression in heart tissue. Therefore, we performed a proteomics study to measure absolute levels of key proteins involved in EETs synthesis and degradation in control and diseased human ventricular heart tissues. Specifically, we measured CYP2J2, POR, sEH, cPLA_2_𝑎, HMOX1, and LPCAT1 in ventricular tissue from CVD patients vs. controls. We also measured total *cis*- and *trans*-EETs (free and esterified) in the same cardiac tissue homogenates. Finally, we determined CYP2J2 activity in the same homogenate, using terfenadine hydroxylation as the in vitro probe reaction [24].

## 2. Results

### 2.1. Total Free and Esterified EETs and DHETs from Human Cardiac Tissue

Control cardiac tissue (*n* = 17) had significantly higher *cis*- and *trans*-EETs relative to diseased tissue (*n* = 24) (Figure 2 and Figure 3). Interestingly, diseased cardiac tissue had significantly lower *trans*-EET levels for all regioisomers compared to control cardiac tissue (Figure 3). The measured *cis*-EET levels and the peak height ratios of both *cis*- and *trans*-EETs from these cardiac tissues are summarized in Table 1 and Table 2, respectively. Total levels of *trans*-EETs in control cardiac tissue were slightly higher than their corresponding *cis*-EETs (Figure 4). Mixed trends were observed when analyzing total *cis*-EETs relative to total *trans*-EETs in diseased tissue (Figure 5). Similar trends were also observed in the DHET levels. Diseased cardiac tissues have significantly less DHETs than control cardiac tissues, except for 5,6-DHET (Figure 6A–E). The ratio of total EET to total DHET levels did not change between control and diseased tissues (Figure 6F).

### 2.2. Protein Quantitation in Human Cardiac Tissue

Protein levels of cPLA_2_𝑎, Lp-PLA_2_, LPCAT1, and HMOX1 were below the limit of detection for most samples and could not be analyzed further. CYP2J2, POR, and EPHX2 levels allowed for absolute quantitation. In general, control cardiac tissues exhibited markedly higher protein levels compared to diseased tissue (Figure 7). CYP2J2 protein levels were lower in diseased cardiac tissue (0.14–0.55 pmol/mg of membrane fraction) compared to control (0.19–0.80 pmol/mg of membrane fraction) (Figure 7A). POR levels were higher in control versus diseased cardiac tissue (Figure 7B). Levels of EPHX2 in control cardiac tissue were significantly higher than in diseased cardiac tissue (Figure 7C). In control cardiac tissue, the ratio of CYP2J2 to POR was approximately 1:12 and averaged 1:13 in diseased cardiac tissue.

### 2.3. CYP2J2 Specific Activity in Human Cardiac Tissue Using Terfenadine as a Probe Substrate

CYP2J2 in cardiac tissue oxidized terfenadine to the alcohol and acid metabolites. Formation of azacyclonol, a CYP3A4/5 specific metabolite was not observed. In addition to the interindividual variability, the type and progression of CVD in the analyzed cardiac tissue may contribute to the extensive variability observed in the data. Control cardiac tissue exhibited significantly higher CYP2J2 activity compared to diseased tissues, as demonstrated in Figure 8.

## 3. Materials and Methods

### 3.1. Chemical and Reagents

EDTA-free Halt protease inhibitor cocktail, dithiothreitol, iodoacetamide, Acros Organics’ ammonium bicarbonate buffer (98% purity), and peptides of interest listed in Appendix A were obtained from Thermo Fisher Scientific (Waltham, MA, USA) and used without further purification. Dulbecco’s phosphate-buffered saline, pH 7.4 (DPBS, 10×), Pierce bicinchoninic acid (BCA) protein assay kit, sodium chloride, potassium phosphate, ammonium acetate, ACS-grade ethyl acetate, chloroform, optima-grade acetonitrile, water, methanol, formic acid, and acetic acid were purchased from Fisher Scientific (Hampton, NH, USA). Terfenadine, hydroxy terfenadine, carboxy terfenadine hydrochloride, triphenylphosphine, azacyclonol, NADPH, and PLA_2_ from *Naja mossambica mossambica* were obtained from Sigma-Aldrich (St. Louis, MO, USA). A 1 mg/mL ethanolic solution of midazolam was purchased from Cerilliant Corporation (Round Rock, TX, USA) and used as an internal standard. Precellys 24 homogenizer, reinforced homogenization tubes and stainless-steel beads were obtained from Bertin Instruments (Rockville, MD, USA). Human serum albumin (HSA) and bovine serum albumin (BSA) were purchased from Calbiochem (Billerica, MA, USA). 14,15-EET and deuterated 14,15-EET, 11,12-EET, 8,9-EET and deuterated 8,9-EET, 5,6-EET and deuterated 5,6-EET, and 14,15-DHET-d_11_ were obtained from Cayman Chemical (Ann Arbor, MI, USA).

### 3.2. Study Samples

Diseased human heart tissues (*n* = 24, 6 from heart transplant patients and 18 from LVAD patients) were collected from the University of Washington Medical Center as surgical excess tissue during cardiac transplants and other procedures by Dr. Jean C. Dinh. The patients’ age ranged between 41 and 70 years old. Left ventricular tissue sections of both control and diseased tissues were immediately flash-frozen in liquid nitrogen and stored at −80°until further processing. Control human heart tissues (*n* = 17) were a gift from BioIVT (Baltimore, MD, USA). The control subjects were between 65 and 78 years old and have been used to quantify other drug metabolizing enzymes in the heart [25].

### 3.3. Quantitation of Total Free and Esterified EETs from Human Cardiac Tissue

The preparation of the cardiac tissue began the day prior to extraction. Ventricular cardiac tissue was patted dry, weighed, minced, and froze in reinforced wall homogenization tubes containing six stainless-steel beads. The following day, homogenization of the cardiac tissue was initiated by adding 0.5 mL of cold 1 × DPBS in the presence of 12.1 μM 4-[[*trans*-4-[[(tricyclo [3.3.1.1^3,7^]dec-1-ylamino) carbonyl] amino] cyclohexyl] oxy]- benzoic acid (*trans*-AUCB), a soluble epoxide hydrolase inhibitor, using Precellys 24 homogenizer at 6800 rpm for 6 × 30 s with 60 s delay between cycles. Ten μL of internal standards (a mixture of 14,15-EET-d_11_, 8,9-EET-d_11_, 5,6-EET-d_11_, and 14,15-DHET-d_11_, 3 μg/mL each) were added to each sample of the homogenized tissue followed by dividing the homogenized tissue into duplicate aliquots of 150 μL. 1 × EDTA-free Halt cocktail protease inhibitor (1:600 *v*/*v*) was added to the remaining heart homogenate before further processing for proteomics analyses. EETs extraction from homogenized cardiac tissue followed the protocol published in [26]. Extracted dried samples were reconstituted in 50 μL solution containing 50% water and 50% of 80:20 acetonitrile: methanol prior to mass spectrometric analysis.

### 3.4. Absolute and Relative Protein Quantitation

Absolute protein quantitation for CYP2J2, EPHX2, HMOX1, and POR and relative protein quantitation for LPCAT1, cPLA_2_𝑎, and Lp-PLA_2_ was carried out using a validated mass spectrometric proteomics method at Department of Pharmaceutics, University of Washington. Twenty-one diseased human heart tissues (five tissues were obtained from heart transplant patients and sixteen tissues were procured from LVAD patients) were analyzed due to limited amount of the samples. The absolute amount of CYP2J2 in one of the diseased samples was not detected, therefore, the sample size of diseased human heart tissue was 20. At least one surrogate peptide for each protein was used for protein quantitation and their corresponding stable labelled peptides with [^13^C_6_, ^15^N_4_]-arginine or with [^13^C_6_, ^15^N_2_]-lysine residues on their C-terminus was used as an internal standard. Peptides used for the protein quantitation are listed in Appendix A.

### 3.5. Protein Extraction and Trypsin Digestion

Proteins from heart homogenates were solubilized, enriched, and extracted using Thermo Fisher Mem-PER^TM^ Plus membrane protein extraction kit following the manufacturer’s protocol. Membrane-bound proteins such as CYP2J2, POR, LPCAT1, and HMOX1 were analyzed using the pellet fraction while cytosolic proteins like EPHX2, cPLA_2_𝑎 were analyzed using the supernatant fraction. Prior to samples preparations for proteomics analysis, total protein content was determined using BCA assay following the manufacturer’s protocol. Sample preparation for proteomic analysis followed the protocol published by Xu et al. [27]. Briefly, heart homogenates were diluted to 2 mg/mL total protein content. Microsomal and cytosolic proteins along with 0.7 mg/mL HSA, 2.7 μg/mL BSA were denatured and reduced with 20 mM ammonium bicarbonate buffer, pH 7.8 and 17 mM dithiothreitol at 95 °C for 10 min with gentle shaking at 300 rpm. After cooling to room temperature in the dark for 10 min, alkylation of denatured proteins was carried out by adding 59 mM of iodoacetamide and incubating the samples at room temperature in the dark for 30 min. Ice cold methanol, chloroform, and water (5:1:4 ratio) were then added to each sample. After mixing and centrifugation at 16,000× *g* for 5 min at 4 °C, the upper and lower layers were removed, and the remaining pellets were dried at room temperature for 10 min. Pellets were washed with ice cold methanol followed by centrifugation at 8000× *g* for 5 min at 4 °C. After removal of the supernatant, the pellets were dried at room temperature for 30 min followed by resuspension in 60 μL of 50 mM ammonium bicarbonate buffer, pH 7.8. To initiate trypsin digestion, 0.04 μg of trypsin (desired protein:trypsin ratio was between 1:10 and 1:100) were added to each resuspended pellet and incubated at 37 °C for 18 h for membrane protein or 16 h for cytosolic protein with gentle mixing at 300 rpm. The reaction was quenched by flash freezing the samples in dry ice. Subsequently, 20 μL of a cold cocktail of stable labelled peptide internal standard (prepared in 80% acetonitrile containing 0.1% formic acid) and 10 μL of cold 80% acetonitrile containing 0.5% formic acid were added to the sample. The samples were vortexed and centrifuged at 4000× *g* for 5 min at 4 °C. The supernatants were collected and analyzed by mass spectrometry.

### 3.6. Mass Spectrometric Assay for Protein Quantitation

The mass spectrometric assay to quantify protein was performed following protocols by Xu et al. [27]. Briefly, samples were analyzed on an AB SCIEX Triple Quadrupole 6500 (PE SCIEX, Concord, ON, CA) coupled to a Waters Acquity UPLC I-class (Waters Technologies, Milford, MA, USA) in electrospray positive ionization mode. Mass spectrometer parameters for each protein of interest are summarized in Appendix A. A Waters Acquity UPLC HSS T3, 1.8 μm, C18, 100 Å, 2.1 × 100 mm column attached to a Phenomenex C18, 2 × 4 mm guard column was used to separate the different peptides with a gradient listed in Appendix A. The mobile phase for this assay consisted of water containing 0.1% formic acid and acetonitrile containing 0.1% formic acid.

### 3.7. Metabolic Activity of CYP2J2 in Heart Tissue

The remaining heart homogenates from total protein quantitation were subsequently used to determine CYP2J2 catalytic activity. Due to limited amount of the heart homogenates, nineteen heart homogenates were used for the metabolic activity assay. Metabolic incubations were performed using 1 mg/mL total heart homogenate protein in the presence of 8 μM terfenadine in 100 mM potassium phosphate buffer, pH 7.4 (final volume 100 μL). Following addition of substrate, the samples were pre-equilibrated for 3 min at 37 °C after which the reaction was initiated with 1 mM NADPH. After 30 min, reactions were terminated with 100 μL of ice-cold acetonitrile containing internal standard (25 nM midazolam). The quenched reactions were vortexed followed by centrifugation at 3500 rpm at 4 °C for 10 min. The supernatant was transferred to a 96-well plate and analyzed using LC-MS/MS.

### 3.8. LC-MS/MS Analysis of Terfenadine Oxidation in Heart Tissue

The terfenadine oxidation mass spectrometric assay was modified from the previous assay developed in our lab [28]. Measurements of terfenadine metabolites were performed on AB SCIEX Triple Quadrupole 6500 (PE SCIEX, Concord, ON, CA) coupled to a Waters Acquity UPLC, I-class (Waters Technologies, Milford, MA, USA). Samples were analyzed in electrospray positive ionization mode. Hydroxy terfenadine, carboxy terfenadine, azaclyclonol, and midazolam (internal standard) were separated on an Agilent Zorbax XDB C8, 5 μm, 2.1 × 50 mm column attached to a Phenomenex SecurityGuard^TM^ HPLC column cartridge. The mobile phase consisted of 10 mM ammonium acetate, pH 5.5 (solvent A) and 10 mM ammonium acetate in 90:10 acetonitrile: water (solvent B). Midazolam and metabolites of terfenadine were separated using the following gradient: solvent B held at 20% from 0 to 1 min, then held at 100% from 2.5 to 4.5 min, followed by re-equilibration to 20% from 4.6 to 6.5 min. Flow rate was constant at 0.35 mL/min throughout the run. For the first one minute of runtime, 100% of the flow was diverted into waste. The mass spectrometer conditions are summarized in Appendix A. In order to prevent carry-over between samples, the injection needle was washed for 10 s with 0.1% formic acid in acetonitrile after each sample injection.

### 3.9. Data Analysis

Mass spectrometry data for EET and DHET quantitation were analyzed using MassLynx 4.1 software while proteomics and terfenadine oxidation data were analyzed using Analyst 1.6.2.

The amount of *cis*-EETs were quantified using an established calibration curve that ranged between 0 to 500 ng/mL. Due to unavailability of commercial neat standards for *trans*-EETs, the measurement of *trans*-EETs were presented as peak height ratio of *trans*-EETs to deuterated *cis*-EETs. The DHETs were measured qualitatively by taking the peak height ratio to 14,15-DHET-d_11_ internal standard.

Absolute quantitation MS-based proteomics was achieved by establishing calibration curves using non-labeled peptides for proteins of interest. In contrast, relative quantitation was determined by compared a protein level of an individual sample to a pooled sample of all the subjects as detailed by Prasad et al. [29].

Terfenadine turnover was determined by adding hydroxy and carboxy terfenadine metabolites in all samples that were supplemented with NADPH. Ratio of each analyte to midazolam was calculated using the peak area ratio of samples with NADPH which was then subtracted from its corresponding peak area ratio of no NADPH control followed by summation of normalized peak area ratio of both terfenadine metabolites.

### 3.10. Statistical Analysis

Statistical analyses were performed using Prism 9.1.0 (GraphPad, La Jolla, CA, USA). All data sets were tested for normality using D’ Agostino & Pearson test and Anderson-Darling test. Based on the results of the normality tests, parametric and non-parametric tests were used to detect differences between groups. Mann–Whitney tests were used to compare *cis*-EET and *trans*-EET levels, DHET levels, and total of terfenadine metabolites between control and diseased tissue. A Wilcoxon matched-pairs signed-rank tests was used to compare the total *cis*-EETs and total *trans*-EETs within the sample. Lastly, the comparison between CYP2J2, POR, and sEH levels in control and diseased tissue were analyzed using unpaired *t*-tests with Welch’s corrections.

## 4. Discussion

This is the first report demonstrating that cardiac disease alters EET levels in human ventricular heart tissue and is associated with differential expression of proteins involved in their formation and degradation including CYP2J2, POR, and sEH. In addition, we report a correlation between EET levels and CYP2J2 activity using terfenadine as a probe substrate.

We previously demonstrated that CYP2J2 metabolizes AA to four regioisomers of *cis*-EETs and that *trans*-EETs are not formed by CYP2J2 [26]. Instead, *trans*-EETs are preferentially formed by ROS and are also found esterified at the *sn*-2 position of phospholipids and therefore, will be released by PLA_2_. In this study, we measured both *cis*- and *trans*-EETs in cardiac tissue. We expected to observe lower *cis*-EET levels in chronically diseased cardiac tissues. We need to be cautious and take into consideration the type and progression of CVD in tissues used in this study when interpreting the results. In diseased heart tissue, both *cis*- and *trans*-EETs are lower than control tissue (Figure 2 and Figure 3). Although measured in two different compartments, our finding of lower cardiac *cis*-EET levels in subjects with heart disease are consistent with observations by Oni-Orisan et al., who reported significantly lower total plasma *cis*-EETs in CAD patients compared to healthy volunteers [15]. An interesting observation is that *trans*-EET were also lower in diseased cardiac tissue compared to control (Figure 3). A potential explanation is that we obtained the diseased ventricular tissues from LVAD or heart transplant patients. These patients suffered from chronic heart disease and were at end-stage heart failure. A study by Saraste et al. demonstrated that the number of apoptotic cardiomyocytes significantly increased in diseased cardiac tissue [30]. In the same study, they also showed the number of apoptotic cardiomyocytes increased significantly with progression of CVD that affected the mechanical, electrical or morphological dysfunction of the myocardium [30]. Since cardiomyocytes make approximately 70–80% of heart tissue and 20–40% of cardiac tissue cellular volume is accounted for by mitochondria, the cumulative apoptotic cardiomyocytes in diseased tissue could be expected to produce less ROS [31,32], consistent with our observation of lower *trans*-EET levels in diseased cardiac tissue.

When closely examining levels of *cis*- and *trans*-EETs in different groups, we noted significantly lower *cis*-EETs compared to *trans*-EETs in control cardiac tissues (Figure 4). An increase in free radical oxidation, which is associated with pathology of many diseases, can lead to an increase of predominantly *trans*-EETs [26]. However, one factor that is easily overlooked in this case is the subjects’ age. Approximately 90% of our cardiac tissue are from elderly patients. Cardiac aging, which is linked to mitochondrial aging, ref. [31] can be associated with an increase in ROS production and reduced mitochondrial respiratory capacity as indicated by a reduction in the phosphocreatine recovery time [33]. This elevated production of ROS can potentially lead to an increase in production of *trans*-EETs relative to *cis*-EETs in the control cardiac tissues that we observed in Figure 4. Unfortunately, we could not measure ROS levels in this study and the link between ROS levels and *trans*-EET levels cannot be verified. In diseased tissue, *cis*-EET levels are not significantly different than *trans*-EET levels (Figure 5). An earlier study in human cardiac tissue showed that control cardiac tissue possessed a lower number of apoptotic cardiomyocytes than diseased cardiac tissue [30]. Therefore, it is plausible that in diseased cardiac tissue, where sections of the hearts were dysfunctional, ROS production during mitochondrial respiration could be reduced. With reduced ROS production, it is anticipated that formation of *trans*-EETs, which are mediated by free radical oxidation, would also be lower [26].

Although not statistically significant, control tissue generally exhibited higher CYP2J2 levels compared to diseased tissue (Figure 7A) using absolute protein quantitation. Since disease state can affect both protein levels and enzymatic activity, we sought to determine CYP2J2 activity in the same heart homogenates used for protein quantitation and EET analysis using terfenadine as a probe substrate. Higher EETs were associated with higher CYP2J2 activity in control tissue vs. diseased (Figure 8). To our knowledge, this is the first report of terfenadine, or any drug, metabolism in control and diseased cardiac tissue. It is important to note that azacyclonol, the major metabolite formed by CYP3A4/5 was not observed in our study confirming that the major metabolic activity of cardiac tissue is attributed to CYP2J2.

In untreated rat liver microsomes, the ratio of total CYPs to POR is reported to be approximately 23:1 [34]. In a different study using rat liver microsomes, Reed et al. immunochemically measured the ratio of total CYPs to POR to be about 5:1 [35]. In control human cardiac tissue, the ratio of CYP2J2 to POR was approximately 1:12 similar to the ratio in diseased cardiac tissue (1:13). Endogenous functions of POR in cardiac tissue remain largely unknown. Loss of POR in mouse was shown to be lethal during embryogenesis [36]. However, conditional knockout of cardiomyocyte-specific POR in mice was not lethal [37]. Because of the maintenance of POR levels in cardiac tissue, it is possible that POR possesses important roles in cardiac homeostasis in addition to supporting CYP2J2 metabolism.

Examining sEH levels, control cardiac tissue expressed significantly higher sEH (encoded by *EPHX2*) than diseased tissue (Figure 8C) which is reflected in the DHET levels (Figure 6A–E). In agreement with the postulated cardioprotective function of EETs, lower sEH protein in diseased cardiac tissue is expected to lead to lower degradation of free EETs prolonging their protective effects. We observed that DHET levels were lower in diseased tissue than controls and the ratio of total EETs to DHETs did not change in control and diseased tissue. However, the observed differences in the DHET levels are largely driven by lower available EETs to be hydrolyzed by sEH. Due to tissue availability and mass spectrometric sensitivity, we could not distinguish the difference between free and esterified EETs and DHETs although we used a potent sEH inhibitor during tissue processing to minimize conversion of EETs to DHETs.

Unfortunately, quantitation of several proteins involved in formation and degradation of EETs, such as cPLA_2_𝑎, Lp-PLA_2_, and LPCAT1 and a stress-marker protein, HMOX1 were not successful due to levels falling below the limit of detection. Furthermore, the native structures of these proteins need to be considered carefully for mass spectral determination. If post-translational modifications are present in these proteins, especially in the peptides used for quantitation, then such modifications could alter protein yield during the protein isolation step and prevent detection in mass spec assays. To achieve optimal results, the protein isolation steps need to be optimized for each protein.

This study had several limitations. Notably, there was no information on the disease state of each heart tissue and our assumption that all auxiliary enzymes involved in biosynthesis and biodegradation of EETs were not changing with disease state could not be verified. In terms of disease progression, we assumed they were at or close to end-stage since patients were undergoing a heart transplant or an LVAD. However, more information regarding the severity of CVD progression could potentially explain some of the variability observed in both EET and protein levels. Another limitation is our inability to verify if the drugs patients were taking affect CYP2J2 activity. A previous study demonstrated that the expression of CYP2J2 was not affected by certain drug treatment in cardiac cells [28], however, not all drugs patients were receiving were tested. Finally, while the changes in *trans*-EETs are very interesting, and their levels seem to vary with disease state, further work that includes measurement of apoptotic cardiomyocytes in the myocardium and ROS levels is necessary to determine the role *trans*-EETs play in cardiac health and disease.

## 5. Conclusions

In general, there were significantly lower *cis*-EETs in chronically diseased heart tissue compared to controls. Significantly lower *trans*-EET levels in diseased cardiac tissue relative to control tissue were observed likely due to cumulative apoptotic cardiomyocytes that lead to an increasing loss of mitochondria. Mitochondria are the main source of cellular ROS [38]. Quantitation of proteins involved in biosynthesis and biodegradation of EETs demonstrates that control tissue generally had significantly higher protein levels of POR and sEH than in diseased tissues. The role of POR in altering EET levels remains unclear given that it is expressed at higher levels than CYP2J2 and therefore unlikely to be the limiting factor in CYP2J2 catalytic turnover. We expected lower sEH protein levels in diseased cardiac tissue to prevent further hydrolysis of EETs and to prolong their cardioprotective effect and our data is supportive of this hypothesis [39]. Therefore, we conclude that the alteration in key proteins involved in biosynthesis and degradation of EETs is in part a result of CVD. We cannot rule out if lower EETs result in cardiac disease and future studies measuring EET levels in patients at different progression stages of CVD will be helpful in determining if EETs precede CVD events.

## Figures and Tables

**Figure 1 ijms-23-12433-f001:**
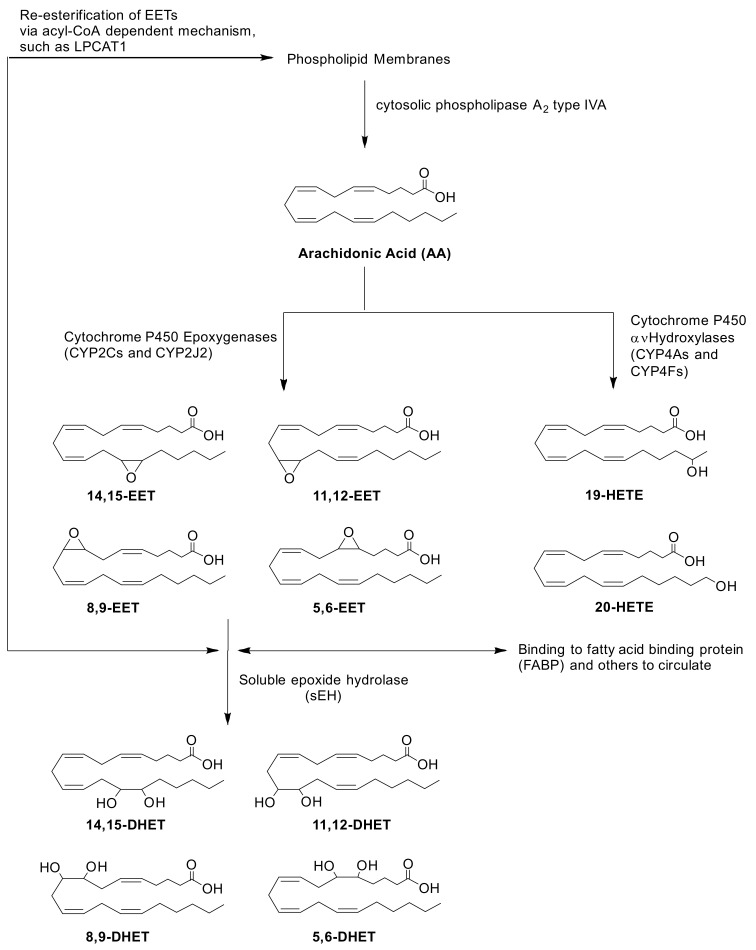
Biosynthesis and biodegradation of EETs via CYP pathways. AA is hydrolyzed from the phospholipid membrane upon activation of cPLA_2_𝑎 (encoded by *PLA2G4A*). Oxidation of AA by CYPs require NADPH-CYP Oxidoreductase (*POR*) to transfer electrons and initiate their catalytic cycle. CYP epoxygenases, CYP2Cs and CYP2J2, mediate the biosynthesis of four regioisomers of *cis*-EETs. Once formed, EETs can either be incorporated back into the phospholipid membrane via acyl-CoA such as lysophosphatidylcholine acyltransferase 1 (*LPCAT1*), bound to other proteins in circulation or hydrolyzed by sEH (encoded by *EPHX2*) to generate DHETs.

**Figure 2 ijms-23-12433-f002:**
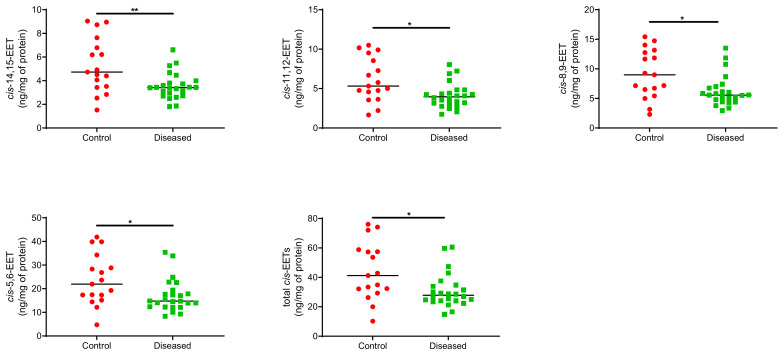
Individual *cis*-EET and total *cis*-EETs extracted from control (*n* = 17) and diseased (*n* = 24) human cardiac tissues. ** indicates *p* ≤ 0.01 and * indicates *p* ≤ 0.05.

**Figure 3 ijms-23-12433-f003:**
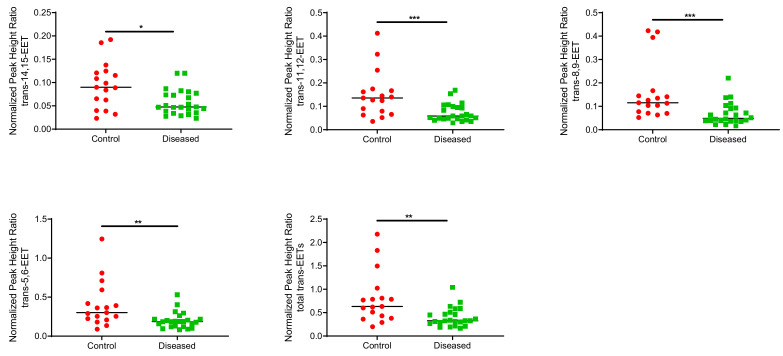
Normalized peak heigh ratio of each *trans*-regioisomer of EETs and total *trans*-EETs extracted from control and diseased human cardiac tissues. *** indicates *p* ≤ 0.001, ** indicates *p* ≤ 0.01, and * indicates *p* ≤ 0.05.

**Figure 4 ijms-23-12433-f004:**
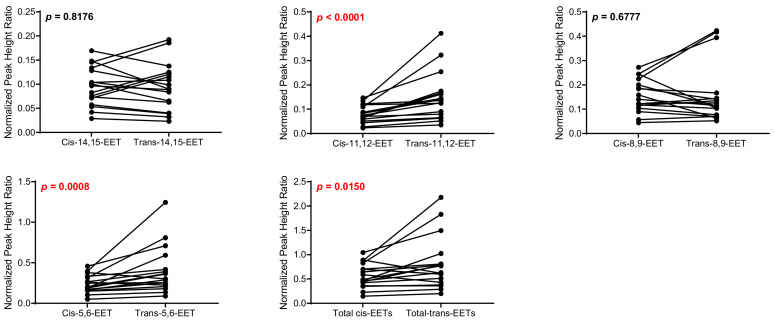
Comparison of total *cis*-EETs and corresponding total *trans*-EETs levels among the different regioisomers in control cardiac tissue. Connecting lines between two points indicate matched pairs of the total *cis*- and *trans*-EETs from the same samples.

**Figure 5 ijms-23-12433-f005:**
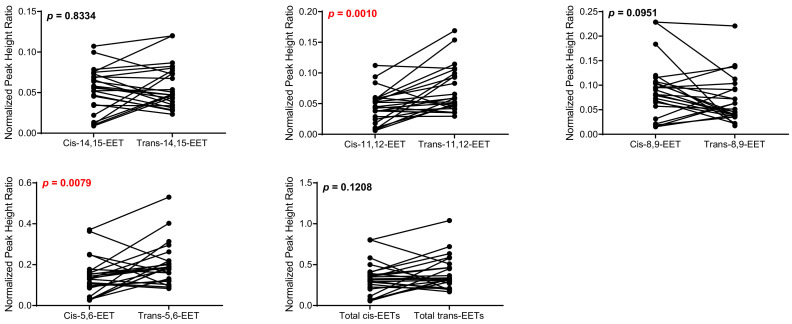
Comparison of total *cis*-EETs and corresponding total *trans*-EETs among the different regioisomers in diseased cardiac tissue. Connected lines between two points indicate matched pairs of total *cis*- and *trans*-EETs from the same samples.

**Figure 6 ijms-23-12433-f006:**
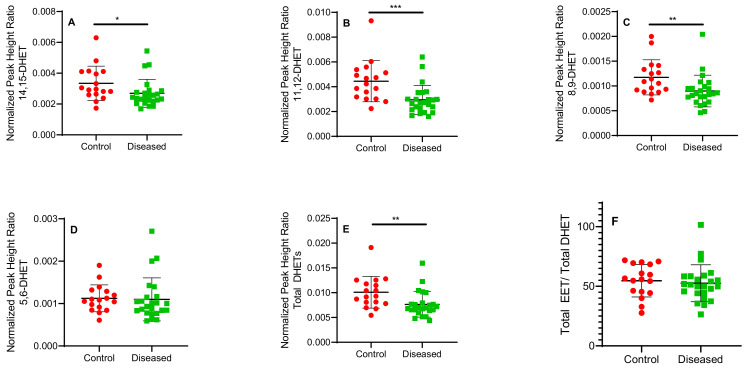
Normalized peak height ratio of each regioisomer of DHETs (**A**–**E**) and ratio of total *cis*-EETs to total DHETs (**F**) extracted from control and diseased human cardiac tissue. *** indicates *p* ≤ 0.001, ** indicates *p* ≤ 0.01 and * indicates *p* ≤ 0.05.

**Figure 7 ijms-23-12433-f007:**
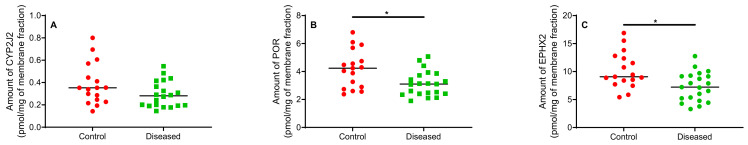
Quantitation of proteins involved in biosynthesis and degradation of EETs. Amounts of (**A**) CYP2J2, (**B**) POR, and (**C**) sEH proteins extracted from human cardiac tissue (*n* = 17 for control tissue and *n* = 20 for diseased tissue). * indicates *p* ≤ 0.05.

**Figure 8 ijms-23-12433-f008:**
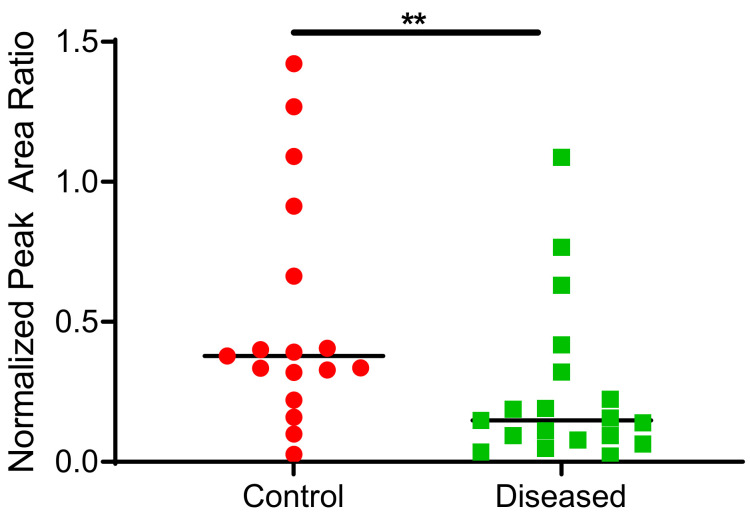
Total terfenadine metabolites (hydroxy + carboxy) formed in control and diseased cardiac tissue homogenates (*n*= 17 for control tissue, *n* = 19 for diseased tissue). ** indicates *p* ≤ 0.01.

**Table 1 ijms-23-12433-t001:** Amount of *cis*-EETs (ng/mg of protein) in control and diseased human heart tissues.

	14,15-EET	11,12-EET	8,9-EET	5,6-EET	Total *cis*-EETs
Control (*n* = 17)	5.29 ± 2.3	6.11 ± 2.8	9.13 ± 4.1	23.7 ± 11	44.3 ± 20
Diseased (*n* = 24)	3.56 ± 1.1	4.12 ± 1.6	6.24 ± 2.6	16.9 ± 6.8	30.9 ± 12

**Table 2 ijms-23-12433-t002:** Peak heigh ratios of *cis*- and *trans*-EETs in control and diseased human heart tissues.

	14,15-EET		11,12-EET		8,9-EET		5,6-EET		Total EETs	
	cis-	trans-	cis-	trans-	cis-	trans-	cis-	trans-	cis-	trans-
Control (*n* = 17)	0.0950 ± 0.040	0.0946 ± 0.05	0.0795 ± 0.046	0.0946 ± 0.05	0.155 ± 0.067	0.160 ± 0.12	0.238 ± 0.11	0.402 ± 0.29	0.567 ± 0.25	0.807 ± 0.55
Diseased (*n* = 24)	0.0533 ± 0.027	0.0575 ± 0.027	0.0453 ± 0.027	0.0725 ± 0.037	0.0906 ± 0.058	0.0683 ± 0.048	0.140 ± 0.093	0.201 ± 0.10	0.330 ± 0.20	0.399 ± 0.20

## Data Availability

All data described are contained within the manuscript.

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
