# Peer review of "Cardiac Disease Alters Myocardial Tissue Levels of Epoxyeicosatrienoic Acids and Key Proteins Involved in Their Biosynthesis and Degradation"

_ijms, 2022, doi:10.3390/ijms232012433_

Round 1

Reviewer 1 Report

This paper deals with the changes in the EET level and CYP2J2 of human heart tissue, the result is interesting. In general, the research work is meaningful, the experimental design is complete and the writing language of this manuscript is also standardized. Therefore, the manuscript can be accepted directly when a few minor problemes were revised.

The author used a sick human heart tissue to quantify other drug metabolic enzymes in the heart. However, the age group is different from the control subject.Will the subclass analysis at the same age group get the same result? As the author says,needed to distinguish from the cardiac aging which is highly identified with mitochondrial aging. 

In this study, protein levels of cPLA2, Lp-PLA2, LPCAT1, and HMOX1 were below the limit of detection for most samples and could not be analyzed. Is this result what you expected? There is insufficient information to support the EET level hypothesis. Describe changes in the CYP2J2-mediated AA metabolism and alter the ratio of cardioprotective EETs.

Author Response

Response to reviewer 1:

Thank you for reading our manuscript and reviewing it, which will help us improve it to a better scientific level.

A major limitation of our study has been the procurement of age-matched control and diseased tissue pairs. Because the process was challenging and time-consuming, we decided to move forward with the tissues that we procured within 5 years.

We attempted to distinguish cardiac aging by examining the level of trans-EETs formed in the tissue. Since the control tissue were collected from older subjects, it was reasonable to expect that the levels of trans-EETs observed in this group are higher than its cis-EETs.

We tried to look for cPLA2 and Lp-PLA2 in ventricular tissue despite suspecting that their levels will be below our quantitation and detection since the Human Protein Atlas reports that there is low protein expression of cPLA2 (encoded by PLA2G4A) and no protein expression of Lp-PLA2 (encoded by PLA2G7) in human heart. We were not sure if the levels will be higher, and perhaps detectable, in the diseased hearts but our study showed they are still undetectable in diseased ventricular tissues. The result for LPCAT1, on the other hand, was unexpected. According to the Human Protein Atlas, there is medium expression of LPCAT1 in healthy human heart tissue. Potential explanations for not detecting LPCAT1 could be that expression in the heart is not localized to the left ventricle and the insufficient amount of tissue used for the analysis. Lastly, the undetected level of HMOX1 was also unexpected. Diseased tissues presumably experienced high level of oxidative stress and therefore, we expected measurable levels of HMOX1. However, ventricular heart tissue might not be the best tissue to detect the level of HMOX1 and examining whole tissue homogenate may yield different results.

As for the reviewer’s last comment: “There is insufficient information to support the EET level hypothesis. Describe changes in the CYP2J2-mediated AA metabolism and alter the ratio of cardioprotective EETs.”

We are unsure what information the reviewer is requesting. We report that cardiac disease alters CYP2J2 and EET levels potentially by downregulating sEH and prolonging the positive effects of EETs. Determining an EET to DHET ratio in our study will not provide further information because we measure both the free and the bound fraction and since over 90% of EETs are bound, the changes in ratio will be indicative of rates and mechanism of incorporation rather than physiological function.

Reviewer 2 Report

This research focused on the EETs and key proteins levels in diseased cardiac tissues. And the authors demonstrated lower EETs levels in diseased tissues. The difference of CYP2J2, POR and sEH has been clarified. The study is interesting. However, there are several suggestion and questions:

Q1: Line36, according to the whole article, the EETs level in disease tissue is LOWER than the control group, but the authors said that ‘for increase EET levels in diseased tissues’. Is this a mistake or does it mean something else?

Q2: Line57, I believe that the authors used the wrong punctuation ‘>’.

Q3: Line244-252, the authors claim that lower sEH in diseased myocardial tissue may lead to lower degradation of free EETs. However, the total EETs level in diseased tissues are lower due to the result, the author did not explain the phenomenon, which is different from the expectation. Meanwhile the line249-250 the results and the comparison of free and esterified EETs were not showed.  

Author Response

Response to reviewer 2:

We would like to thank the reviewer for the careful review of this manuscript

Q1: Line 36, the word “increase” is simply a mistake. It is supposed to be “decrease”. It has been corrected in the manuscript.

Q2: The punctuation “>” has been corrected to a period.

Q3: We included the explanation of why the total EET levels were lower in diseased tissues. In lines 246-247, we mentioned that sEH is responsible for degradation of free EETs. The esterified EETs would not be available, and are not substrates to the cytosolic enzyme, to be hydrolyzed by the sEH. That is the reason we speculated that the level of free EETs would be higher in diseased tissues. Unfortunately, we were not able to distinguish between free and esterified EETs or confirm that free EETs are higher in diseased tissue vs. controls because of procured tissue amount and mass spectrometric sensitivity limitations. This limitation has been added in the manuscript (lines 252-253).

Reviewer 3 Report

The study by Aliwarga and co-workers aim to investigate the levels of epoxyeicosatrienoic acids (EET) and their regulatory proteins in healthy and diseased myocardial tissue. The authors found cis- and trans-EET were elevated in controls, compared to diseased tissue, which was associated with increased levels of CYP2J2, POR, and EPHX2. CYP2J2 activity was also elevated in control tissue. While the authors have presented interesting observations, the study has several limitations, and more work is needed to support the authors conclusions.

Major comments:

1.     As the authors pointed out, there is limited information on the background of the diseased tissue – how many tissues were from transplantation donors and how many were from LVAD recipients – are there differences in EET and enzyme levels between these two diseased groups?

2.     The argument of ROS being elevated in control tissue is interesting given the age of the donors. The authors could confirm this by evaluating enzymes that regulate ROS/antioxidant systems.

3.     Similarly, the apoptosis argument in diseased tissues could be further supported by histological evidence.  

Minor comments:

1.     The authors are advised to proofread the manuscript – line 262; Eric’s DMD paper?? Supplementary table 1; “Several proteins missing from title”?? Indeed several proteins from the table are missing in the title.

Author Response

Response to the reviewer:

We would like to thank the reviewer for taking the time to review this manuscript.

Major comments:

  1. There are 6 heart transplant patients and 18 LVAD patients. The LVAD patients are also at end stage disease, but they are awaiting a heart transplant. Based on unpaired non-parameteric t-test analyses of the total cis- and total trans-EETs, there are no significant differences between total cis- and total trans-EETs between tissues obtained from heart transplant and LVAD patients (p = 0.1036 for cis-EETs and p = 0.8712 for trans-EETs).

The comparison of POR and sEH levels between tissues from heart transplant and LVAD patients also do not show any significant difference. There is a significantly higher CYP2J2 level in tissue from heart transplant patients than LVAD patients. However, these statistical analyses are most likely inaccurate due to significant difference in sample size between tissues obtained from heart transplant and LVAD patients. The small sample size of the tissue from heart transplant patients does not have the statistical power to draw any meaningful conclusions. Due to this reason, we combined tissues from heart transplant and LVAD patients into one diseased group.

  1. We attempted to quantify the level of HMOX1 which would be a great marker for oxidative stress that regulate ROS/antioxidant systems. Unfortunately, the protein levels of HMOX1 were below the limit of our quantitation and detection of the mass spectrometer. Ventricular tissue might not be the best tissue to measure the level of HMOX1. Perhaps red blood cells might be a better tissue to monitor HMOX1 level.

  1. Unfortunately, due to limited amount of ventricular tissue that we received, we did not have enough tissue for histology, mass spec analysis and activity assays. However, in our previous publication ((Aliwarga et. al., Biomedicines, 2020, DOI: 3390/biomedicines8060144), we showed that there is more apoptotic cells in wild-type mice compared to cardiomyocyte-specific overexpression CYP2J2 transgenic mice that were subjected to myocardial infarction, indicating that MI increases apoptotic cells and higher EETs resulting from overexpression of CYP2J2 in the mouse heart reduces the number of apoptotic cells.

Minor comment:

  1. Thank you for this comment. It was an oversight, the citation on line 262 has been added and the proteins from the table have been added to the legend of Supplementary Table 1.

Reviewer 4 Report

Dear authors,

I have read your paper, and it is showing interesting results, but in my opinion, it is unacceptable to construct a research paper without a ”Material and Methods” section, with a detailed description of the methods used.

Please construct such a section and I would gladly review again your paper. 

Good luck!

Author Response

Response to the reviewer:

We would like to thank the reviewer for taking the time to review this manuscript. We draw the reviewer’s attention to the “Material and Methods” section. There is a detailed description of the materials and methods used for the study from page 9 to 12. We look forward to additional comments the reviewer may have after reviewing the materials and methods section.

Round 2

Reviewer 3 Report

With regard to statistical power, could the authors kindly explain why only N=20 and N=19 diseased tissue were used in studies represented by Figure 7 and 8, respectively (and not all 24 samples).

Author Response

Reviewer 3, Round 2

With regard to statistical power, could the authors kindly explain why only N=20 and N=19 diseased tissue were used in studies represented by Figure 7 and 8, respectively (and not all 24 samples).

Response

The authors would like to thank the reviewer for taking the time to comment on our manuscript.

For our proteomics analysis (Figure 7), CYP2J2 could not be detected in one of our diseased human heart tissues as mentioned on lines 348-350 (“The absolute amount of CYP2J2 in one of the diseased samples was not detected, therefore, the sample size of diseased human heart tissue was 20.”).

For terfenadine activity analysis (Figure 8), we only used 19 due to limited amount of heart homogenates as mentioned in the manuscript on lines 396-397 (“Due to limited amount of the heart homogenates, nineteen heart homogenates were used for the metabolic activity assay.”).

Reviewer 4 Report

Dear authors, thank you for your work.

I am content with the progress and the corrections. I only have an observation, the Materials and Methods must be the second, after the Introduction, not the fourth. 

Good luck!
